# Microbial thermogenesis is dependent on ATP concentrations and the protein kinases ArcB, GlnL, and YccC

**Puneet Singh Dhatt[1], Stephen Chiu[1], Tae Seok Moon[1,2]***

**1** Department of Energy, Environmental and Chemical Engineering, Washington University in St. Louis, St. Louis, Missouri, United States of America, **2** Division of Biology and Biomedical Sciences, Washington University in St. Louis, St. Louis, Missouri, United States of America

\* tsmoon7@gmail.com

## Abstract

Organisms necessarily release heat energy in their pursuit of survival. This process is known as cellular thermogenesis and is implicated in many processes from cancer metabolism to spontaneous farm fires. However, the molecular basis for this fundamental phenomenon is yet to be elucidated. Here, we show that the major players involved in the cellular thermogenesis of *Escherichia coli* are the protein kinases ArcB, GlnL, and YccC. We also reveal the substrate-level control of adenosine triphosphate (ATP)-driven autophosphorylation that governs cellular thermogenesis. Specifically, through live cell microcalorimetry, we find these regulatory proteins, when knocked out in a model *E. coli* strain, dysregulate cellular thermogenesis. This dysregulation can be seen in an average 25% or greater increase in heat output by these cells. We also discover that both heat output and intracellular ATP levels are maximal during the late log phase of growth. Additionally, we show that microbial thermogenesis can be engineered through overexpressing *glnL*. Our results demonstrate a correlation between ATP concentrations in the cell and a cell's ability to generate excess heat. We expect this work to be the foundation for engineering thermogenically tuned organisms for a variety of applications.

## Introduction

The concepts of life and heat have been considered interdependent since the time of the ancients [1]. Aristotle was the first person in recorded history who formed a theory on the origin of life. In his treatise *Generation of Animals* in the 4th century BCE, he wrote on the importance of a "vital heat" that is responsible for animating organic material into life [2]. Yet, there still remains the underexplained fact that all organisms release heat [3].

Studies on cellular thermogenesis became popular at the end of the 18th century after the first calorimeters were constructed to measure live cell heat generation [4–8]. Adair Crawford's seminal work in 1779 studied the heat output of guinea pigs [9]. Just 1 year later, Lavoisier and Laplace demonstrated the connection between metabolism and thermogenesis using a novel ice-based calorimeter [10,11]. To this day, the leading theory in molecular biology is that

**Data Availability Statement:** Raw data for all figures can be found in the Supporting Information file S1 Data. Gene annotations are available through Metacyc (https://metacyc.org/). All other relevant data are within the paper and its

Supporting Information files. Model development was conducted in MATLAB. All codes for the generation of the MTM model are available on the Microbial Thermogenesis Model GitHub repository: (https://github.com/pdhatt/microbialthermogenesismodel.git). MTMv1.3.1 is archived in Zenodo under 10.5281/zenodo.8400157.

**Funding:** This work was supported by the Office of Naval Research (N00014-21-1-2206 to T.S.M). The funders had no role in study design, data collection and analysis, decision to publish, or preparation of the manuscript.

**Competing interests:** The authors have declared that no competing interests exist.

**Abbreviations:** ATP, adenosine triphosphate; AUC, area under the curve; HK, histidine kinase; MTM, microbial thermogenesis model; RR, response regulator; SD, standard deviation; SEE, standard error of the estimate.

heat is generated at the molecular level through metabolism. In fact, it is said that 50% to 60% of the energy that is stored in metabolic substrates is released as heat [12].

At the beginning of the 20th century, work in cellular thermogenesis was largely focused on studying microorganisms' ability to cause "spontaneous heating" of farm products. A process which, to this day, causes farm fires across the world [13–21]. This work went on to influence cancer research with the establishment of the Warburg Hypothesis in 1924 [22,23]. Recent work in microbial thermogenesis is mainly focused on the use of isothermal microcalorimetry to monitor microbial activities such as the metabolic dynamics of soil bacteria [24,25], marine sedimental heat production [26], and antibiotic resistance behaviors to determine both antibiotic mechanism and optimal dosage [27]. In eukaryotic systems, microcalorimetry has been applied to study cell thermogenesis to understand antiviral responses of pharmaceuticals [28,29], study mitochondrial and endoplasmic reticular functions [30], explain pathogen origins of obesity and anemia [30–33], and explore cancer phenotypes for advanced diagnostics [34,35].

Given that cellular thermogenesis is a fundamental property of organisms, we need to understand its impacts on complex symbiotic systems in nature. One such system is the human microbiome. It has recently been calculated that up to 70% of the energy released by a human each day can be attributed to the human microbiome [36]. Furthermore, microbial heat contribution has been observed in healthy animal models wherein antibiotic treatment lowers average body temperature by an average of 1˚C [37–40]. This implies a key role of the human microbiome in human energy homeostasis. Gut flora have already been proven to be involved in numerous host regulation mechanisms, including endocrine function, [41] genetic expression [42,43], aging [44], and immune processes [45].

To our knowledge, the field of cellular thermogenesis has thus far been strictly observational. To elucidate the important metabolic agents and gain a fundamental mechanistic understanding of microbial thermogenesis, we hypothesized that there are a set of regulatory proteins that determine a cell's thermogenesis. To test this hypothesis, we screened a single-gene knockout library of a model organism, *Escherichia coli* K-12 BW25113 (henceforth referred to as *E. coli* BW25113), for thermogenesis. The functional understanding gained from this work can then be used to engineer new strains with altered thermogenesis and transition the field from observation to creation in the future.

In this work, we identify both a set of proteins that regulate microbial thermogenesis and the key metabolite, adenosine triphosphate (ATP), through which this regulation occurs in *E. coli*. A mathematical model was also developed to predict thermogenesis using ATP as the sole metabolite upon which thermogenesis is dependent. This work will allow us to engineer microbial thermogenesis to develop increased thermogenic strains for many applications in the future.

## Results

### Experiment design using calorimetry

Calorimetric data is recorded using a thermocouple to correlate temperature changes to an electric signal (voltage) that is then calculated as heat energy by computer software. In general, there is a significant trade-off between the number of samples and sensitivity for calorimetry instruments, especially above 8 sample wells [24]. Considering this trade-off, we chose a TAM Air Calorimeter, which has a sensitivity of approximately 1 $\mu$W. The initial seeding density of cells was optimized so that our final maximal signal for heat flow was reached at 5 h. The base-lining was done to settle for 12 h before measurements were taken, such that the throughput of these experiments is heavily limited at just 6 experimental samples, along with 1 negative

control and 1 positive control, tested per experiment. The experimental workflow is summarized in Fig 1.

Given this limitation in throughput, we tested rationally selected mutant groups of a single-gene knockout library [46]. We hypothesized that there is a global regulatory pathway for heat generation. In pursuit of this hypothesis, we tested all 199 transcription factors present in *E. coli* BW25113. We also included the 29 histidine kinases (HKs) and 32 response regulators (RRs), which are often involved in global metabolic regulation through two-component signal transduction pathways [47]. The regulated pathways include carbon metabolism, nitrogen assimilation, and chemotaxis among others [48]. Furthermore, we tested 19 proteins related to central metabolism and 120 other proteins. There are 29 proteins that belong to both the transcription factor and RR groups, meaning 370 unique single-gene knockouts tested in this study.

## Protein-level regulators of thermogenesis

We probed the single-gene knockout space for increased normalized heat generation, which marked thermogenic dysregulation (Fig 2A and S1 Table). Most strains fell within 10% of the wild-type *E. coli* BW25113 heat generation (Fig 2B). Only 3 extremely high performers were considered in this experiment for further exploration. These strains exhibited an approximate 25% or greater increase in thermogenesis over wild type and were selected to be tested in biological triplicate. Most strains failed to maintain this increase threshold after triplicate measurement. However, through this analysis, 3 strains became evident as extreme heat producers. These are Δ*arcB* ($p$ = 0.0031), Δ*glnL* ($p$ = 0.0321), and Δ*yccC* ($p$ = 0.0104). These strains had an average thermogenesis fold increase of 1.32 (Δ*glnL*), 1.30 (Δ*yccC*), and 1.25 (Δ*arcB*) over wild type (Fig 2C). It is worth noting that in every instance, the thermogenesis of these strains was above that of the wild-type strain for the same experiment (S1 Fig).

Interestingly, these 3 genes encode proteins that are all autophosphorylating protein kinases. Specifically, *arcB* and *glnL* are well-studied global regulators of aerobic metabolism and nitrogen assimilation, respectively, and act through a two-component signal transduction pathway [48,49]. These are also 2 of the HKs with the highest phosphorylation rates [48]. Furthermore, since there are 29 HKs and 32 RRs in *E. coli*, some two-component signal transduction pathways have significant crosstalk between HK-RR pairs [47]. However, having screened all the HK and RR genes individually, we minimized the impact of individual pairs' crosstalk variations in our data.

Although the gene *yccC* (*etk*) is an autophosphorylating protein kinase, it is a tyrosine kinase that is not thought to be involved in two-component signaling in *E. coli*. Prokaryotic organisms are not thought to have sensor-protein tyrosine kinases, which are considered unique to eukaryotic organisms [50]. However, *etk* has been implicated in antibiotic resistance and is known to phosphorylate the heat shock sigma factor [51,52].

Additionally, none of the transcription factor knockouts significantly increase cellular thermogenesis (Fig 2A). This result is unexpected as protein kinases generally regulate gene expression using transcription factors. Significantly increased thermogenesis in the protein kinase knockout mutants, but not the knockout mutants of the downstream transcription factor, can be explained in 2 ways. The knockout of a single protein kinase could increase the incidence of crosstalk of HK-RR pairs. Alternatively, it is not the regulatory behavior of these protein kinases that is leading to these strains' thermogenic effects, but instead the substrate-level effects of ATP that are causing this phenomenon of increased heat output. If the latter is the case, then the system would not only collapse to be more engineerable but also have a global regulator for cellular thermogenesis at the metabolite level, since all of the extreme thermogenic strains are the knockout mutants of self-phosphorylating protein kinases.

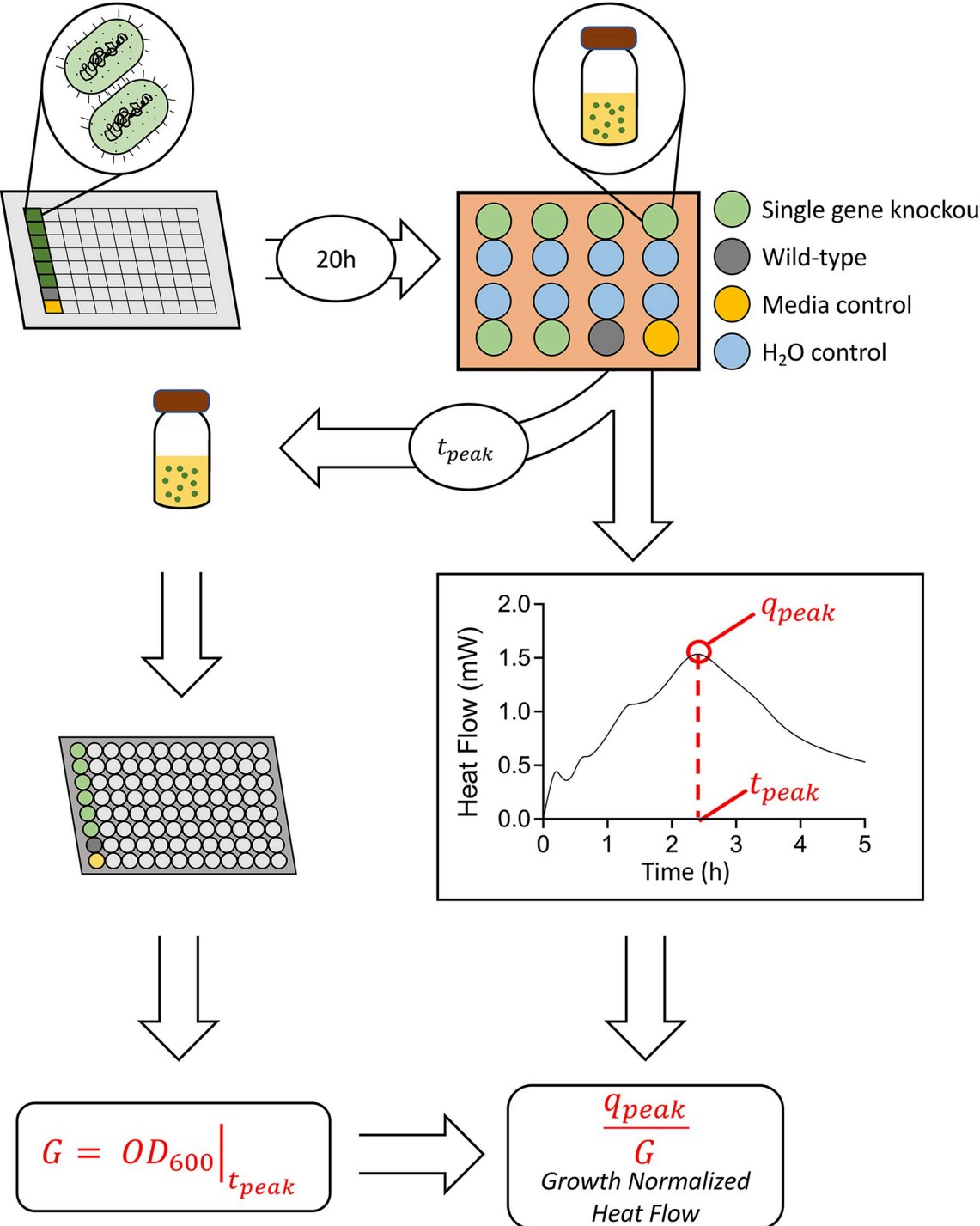

**Fig 1. Methodology of live cell microcalorimetry experiments.** The method by which cells are grown, seeded, and analyzed for their heat generation from the microcalorimeter is shown. First, *E. coli* BW25113 single-gene knockout cells (green), *E. coli* BW25113 wild-type (gray), and a negative control of media are seeded for overnight culture. Then, cells are inoculated at an initial $OD_{600}$ value of 0.005 in a 20 mL glass ampoule. Ampoules are sealed and lowered into the microcalorimeter using $ddH_2O$ as the negative control. After 5 h, the cells are removed from the microcalorimeter, and the final $OD_{600}$ value is measured. The sample heat flow curves are analyzed for the peak value ($q_{peak}$). This value is normalized by the final $OD_{600}$ value to get a growth-normalized heat flow measurement.

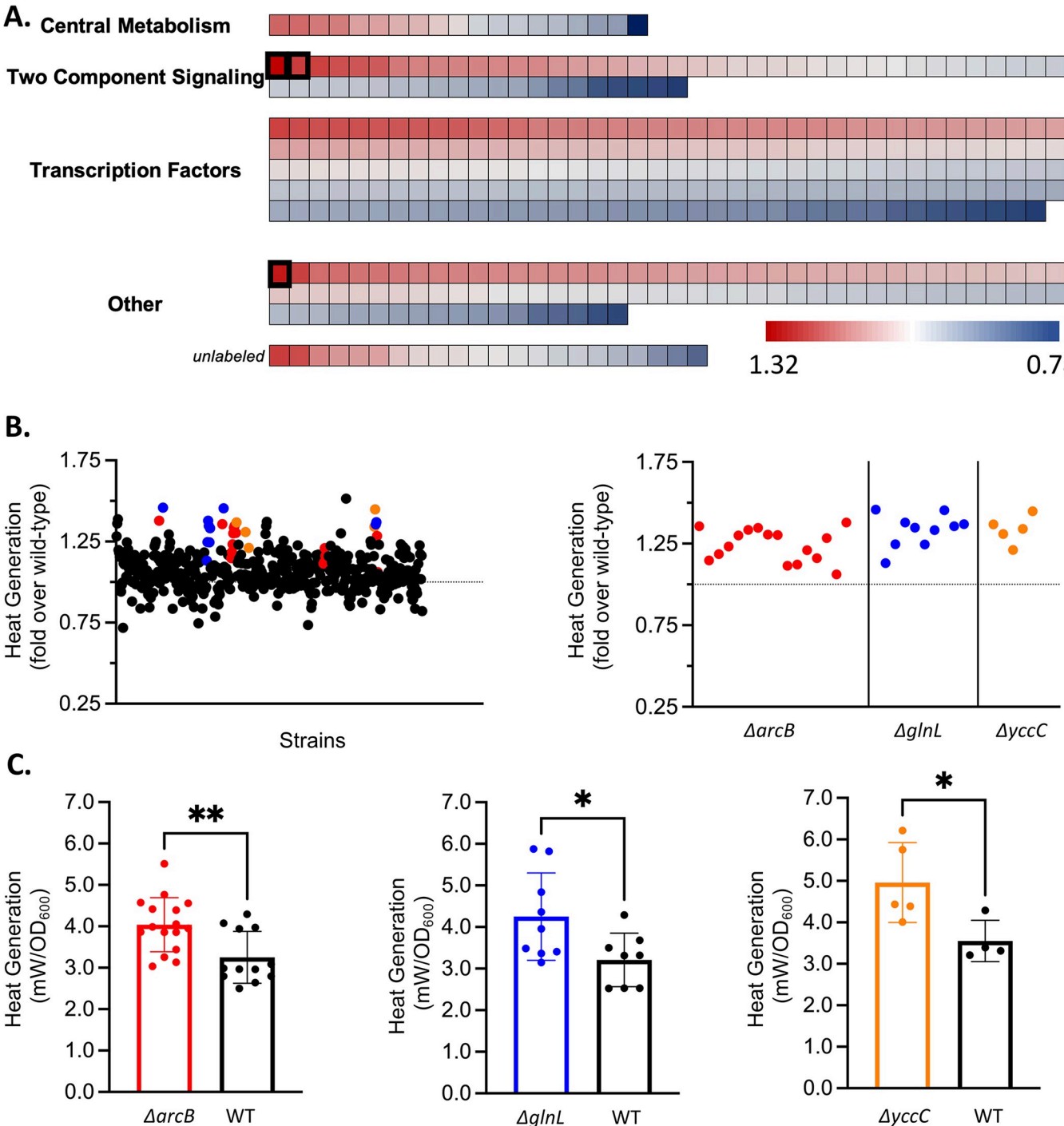

**Fig 2. Heat generation screening of single-gene knockout strains.** (A) Heat map of categorized single-gene knockouts (heat flow fold change over wild-type; $N = 370$). Strains are categorized by their respective knockout gene's function as central metabolic, two-component signal transduction, transcription factors, or other. Other strains either fall into none of the other categories or do not have 3 or 4 letter gene names assigned and are noted as unlabeled. See S1 Table for full data. (B) Scatter plot summarizing all results as fold change over wild type. All tested single-gene knockout strains are shown (black). All data is normalized to wild-type result (dotted line) respective to experiment. $\Delta arcB$, $\Delta glnL$, and $\Delta yccC$ data are visualized separately, as well. (C) Growth-normalized heat flow (mW/OD$_{600}$) bar graph against respective wild-type data (black). High-heat producers highlighted: $\Delta arcB$ ($N = 16$, t = 3.335, df = 21.04)–red, $\Delta glnL$ ($N = 9$, t = 2.435, df = 11.59)–blue, $\Delta yccC$ ($N = 5$, t = 6.723, df = 2.599)–orange, and wild-type–black. Statistical significance by two-tailed T test, *$p < 0.05$, **$p < 0.01$. Error bars: ± SD. Data for all individual replicates can be found in S1 Data.

## Fundamental understanding of cellular thermogenesis at the chemical level

To better understand why a few strains increase in cellular thermogenesis, while most do not, we wanted to explore some of the metabolic realities of extreme thermogenic cells. To do this, we moved forward with our 3 thermogenic strains: Δ*arcB*, Δ*glnL*, and Δ*yccC*. We hypothesized that the intracellular ATP concentration was the key metabolic regulator of this pathway. ATP is one of the highest energy compounds in the cell. It is generated from the catabolism of substrates to drive anabolic reactions [53]. Additionally, $Ca^{2+}$-ATP coupled transporters have been implicated in mitochondrial thermogenesis in mammalian cells [26,54].

We investigated intracellular ATP concentrations in our extreme thermogenic strains as well as a wild-type control over time. We correlated the ATP concentration alongside cellular growth (monitored by $OD_{600}$) and heat flow output. The results show that intracellular ATP concentrations and cellular thermogenesis peak were at the same relative time of the growth curve, the late log phase (Fig 3). The qualitative peak coincidence was confirmed quantitatively by area under the curve (AUC): the maximum integrals of the experimental results' curves were found between hours 3 and 4 of cell growth (S2 Fig). The intracellular ATP concentration could only be calculated after the 2 h time point due to the sensitivity of the assay. The growth rates and curves of all strains are very similar (Fig 3A). Additionally, the general trends observed in the intracellular ATP concentration and the heat generation curves are conserved as well (Fig 3B and 3C).

During the late log phase, the cell is exiting the steady-state growth phase and is transitioning into the stationary phase. This transition is very dynamic and involves the coordinated down-regulation and up-regulation of proteins at the levels of transcription, translation, and protein degradation [55]. This massive shift in metabolism naturally leads to the accumulation of intracellular ATP.

To better understand the link between ATP hydrolysis and microbial thermogenesis, 3 overexpression plasmids were constructed to put *arcB*, *glnL*, and *yccC* (denoted *arcB+*, *glnL+*, and *yccC+*, respectively*)* under anhydrotetracycline-inducible overexpression. These genetic circuits were then transformed into wild-type *E. coli* BW25113. First, the overexpression strains were compared for microbial thermogenesis in the induced and uninduced conditions (Fig 4). The induced *glnL* strain had a significantly decreased, growth-normalized heat generation when compared to the wild-type strain with a *p*-value of 0.0400 (Fig 4A). However, the lack of microbial heat generation and a minimal cell growth after 4 h in the case of the induced *arcB* and *yccC* strains reveals that these kinases' overexpression is lethal (S3 Fig). This was likely caused by the coincubation with the inducer from the beginning of the seed culture, instead of the timely induction at the later growth phase, an experimental limitation necessary to maintain the calorimeter sample sealed.

To evaluate the reason for the observed decrease in cellular thermogenesis due to the overexpression of GlnL, the *glnL* overexpression strain was assayed for intracellular ATP level and growth (Fig 4B and 4C). A coincidence of the peak of intracellular ATP concentration and microbial thermogenesis was partially observed. The quantitative correlation was also confirmed by integration (S4 Fig). The *glnL+* strain also shows an increased lag phase time compared to the wild-type strain (Fig 4B). Additionally, the intracellular ATP level of *glnL+* is significantly reduced compared to that of wild type at the point of maximal thermogenesis (Fig 4B and 4C).

## Development of a microbial thermogenesis model

Hypothesizing that intracellular ATP is the metabolite that determines cell's thermogenesis, we should be able to extract the experimental thermogenesis data from experimentally derived

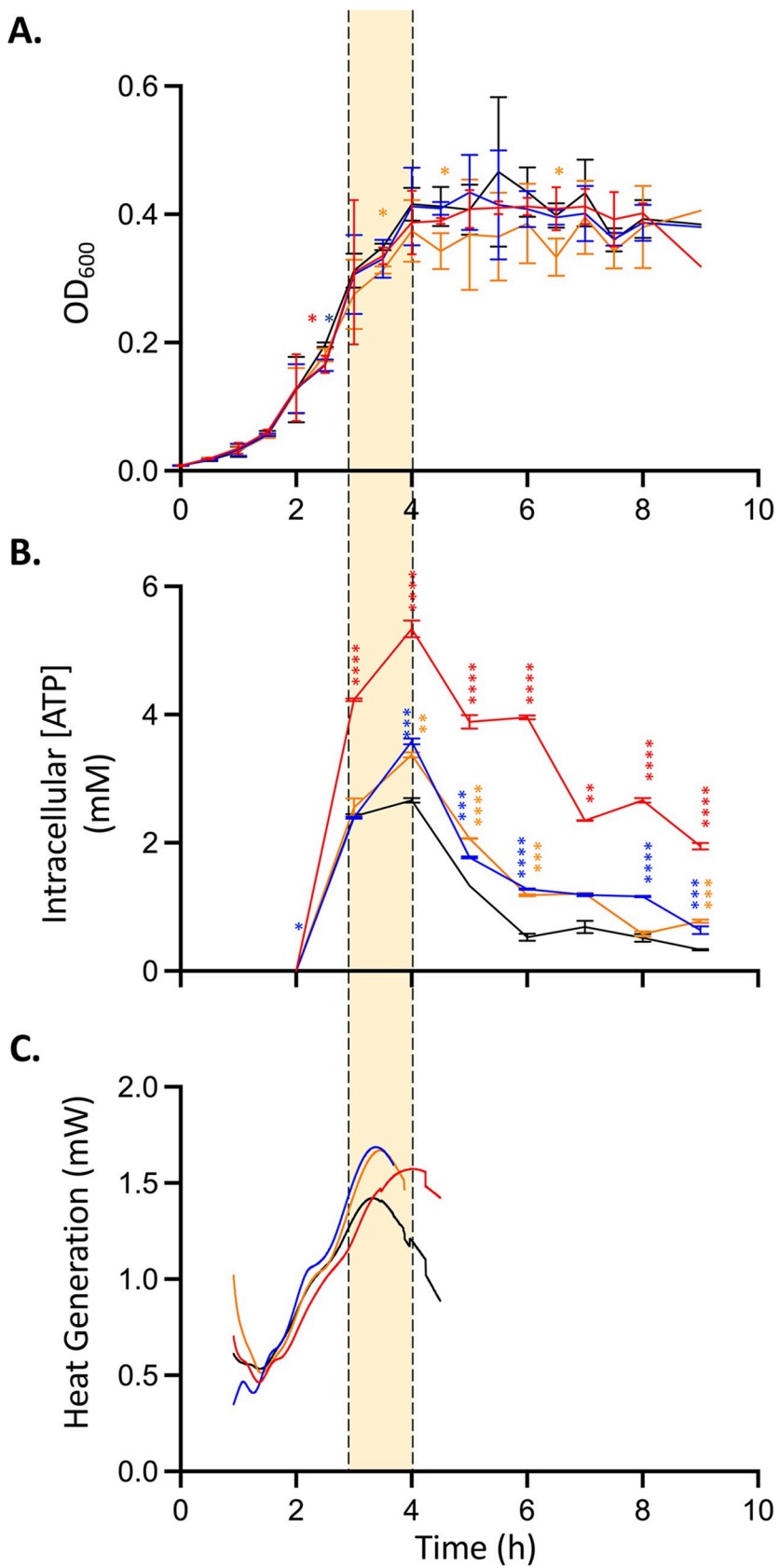

**Fig 3. Microbial growth, intracellular ATP, and thermogenesis assay of selected kinase knockouts.** (A) Average strain growth ($OD_{600}$) curves over time. (B) Average strain intracellular ATP concentration (mM) over time. (C) Average strain heat flow (mW) curves over time. Curve with error bars is included in S11 Fig. *ΔarcB*–red, *ΔglnL*–blue, *ΔyccC*–orange, and wild-type–black. Significance calculated for each strain against wild type by ANOVA. Error bars: ± SD. *$p < 0.1$, **$p < 0.01$, ***$p < 0.001$, ****$p < 0.0001$. Data for all individual replicates can be found in S1 Data.

parameters. Assuming that ATP is the sole, key metabolite responsible for cellular thermogenesis and that the dephosphorylation of ATP releases 7.3 kcal/mol, we can calculate the heat generation from a single cell and extend this to the entire population to model the cellular thermogenesis [56,57]. The inputs in this model are cellular growth and the intracellular ATP concentration, both as analytic functions of time. Cellular growth data was fit using the Gompertz growth equation, while the intracellular ATP concentration was fit using a lognormal distribution (S2 Table). Then, using the constraint-based optimization (COBRA) model iML1515, the steady-state ATP flux in an *E. coli* cell can be calculated [58]. This algorithm defines the microbial thermogenesis model (MTM) (Fig 5A).

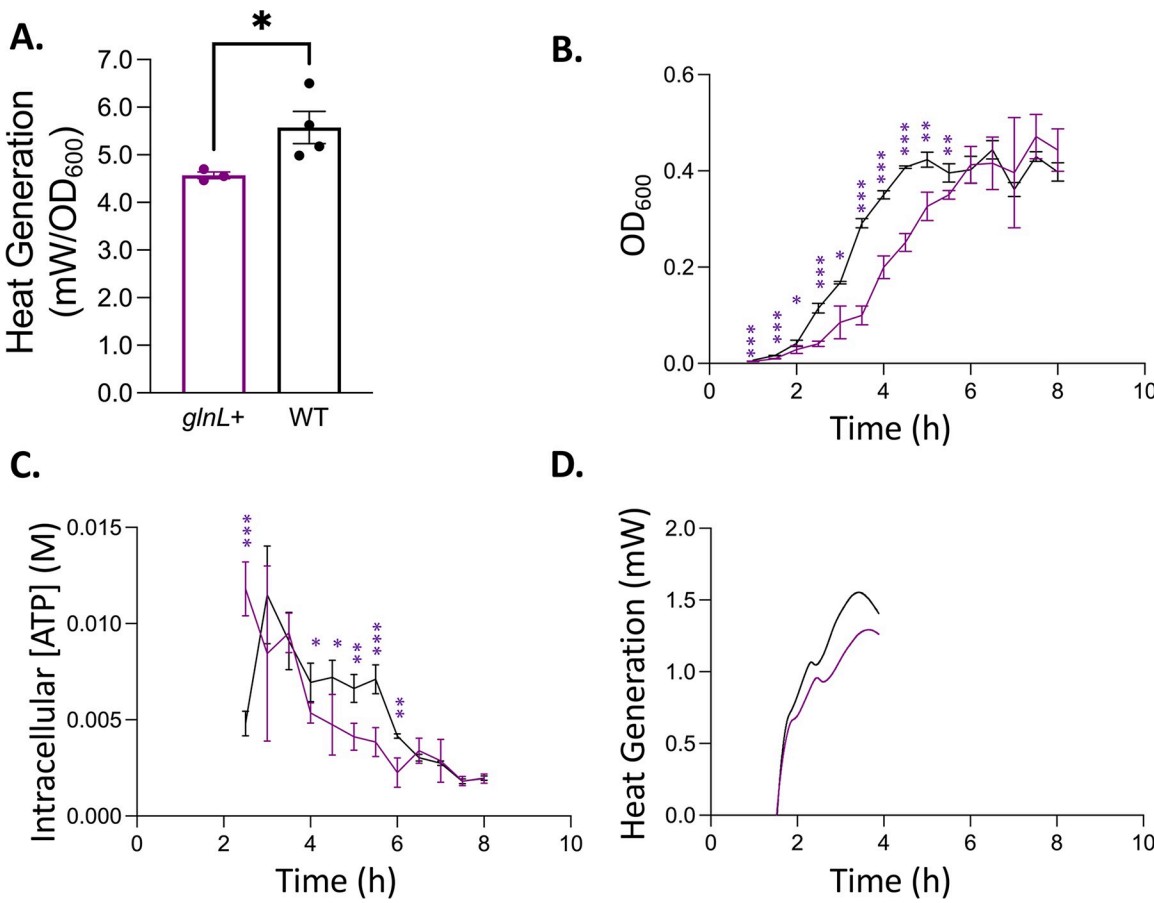

**Fig 4. Effects of *glnL* overexpression on cellular growth, ATP, and thermogenesis.** (A) Growth-normalized heat flow (mW/$OD_{600}$) bar graph of *glnL* overexpression strain (*glnL+*, purple, $N = 3$, $p = 0.0400$) against respective wild-type data (WT, black, $N = 3$). (B) Average strain growth ($OD_{600}$) curves over time. (C) Average strain intracellular ATP concentration (mM) over time. (D) Average strain heat flow (mW) curves over time. Curve with error bars is included in S12 Fig. Significance calculated by Student's *T* test with Welch's correction. Error bars: ± SD. *$p < 0.1$, **$p < 0.01$, ***$p < 0.001$, ****$p < 0.0001$. Data for all individual replicates can be found in S1 Data.

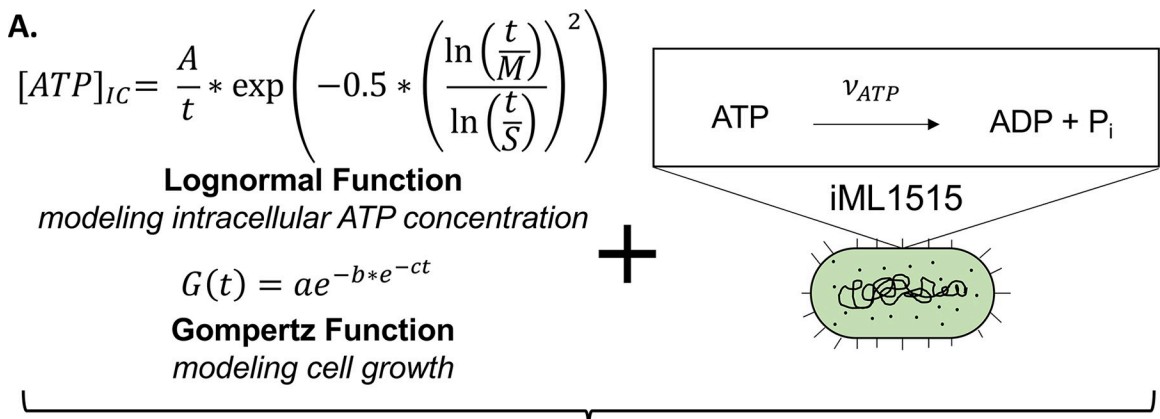

**Microbial Thermogenesis Model**

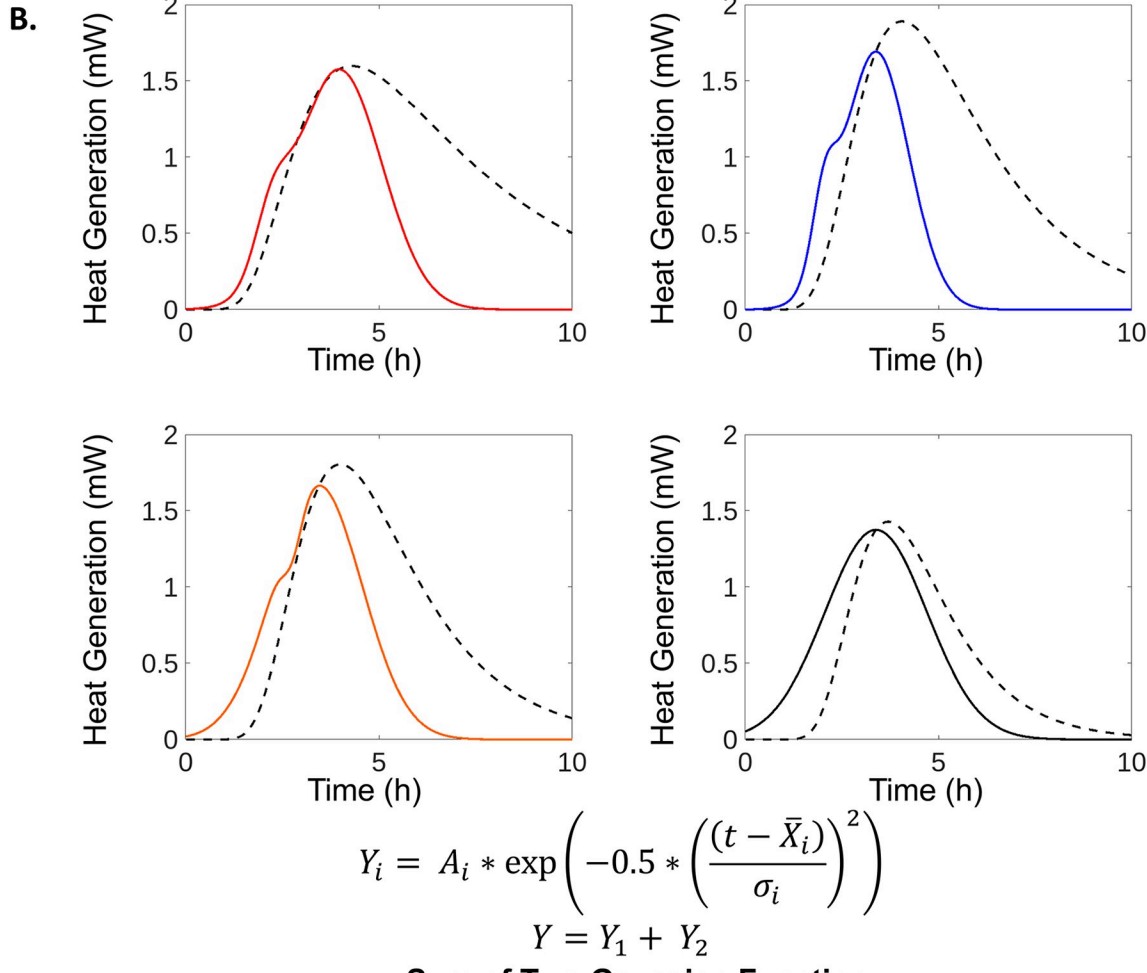

$$Y_i = A_i * \exp\left(-0.5 * \left(\frac{(t - \bar{X}_i)}{\sigma_i}\right)^2\right)$$

$$Y = Y_1 + Y_2$$

**Sum of Two-Gaussian Function**

**Fig 5. MTM analysis.** (A) MTM utilizes the Lognormal and Gompertz functions to model microbial intracellular ATP concentration and growth, respectively. Additionally, the model assumes an intracellular ATP pathway flux from wild-type *E. coli* calculated using iML1515. Gompertz growth equation: a–asymptote as $t \rightarrow \infty$, b–displacement along x-axis, c–growth rate, and t–time. Lognormal equation: M–geometric mean, S–geometric standard deviation, A–factor related to amplitude. (B) Average heat generation is fitted by the sum of 2 Gaussian equations (solid line) and plotted against the respective MTM approximations for each strain's heat generation over time (dashed line). *ΔarcB*–red, *ΔglnL*–blue, *ΔyccC*–orange, and wild-type–black. Sum of 2 ($i$ = 1,2) Gaussian equations: A–amplitude, $\bar{X}_i$–mean, t–time, and $\sigma_i$–standard deviation. Data for all individual replicates can be found in S1 Data. All graphs were generated using MTM MATLAB code.

The MTM matches with the sum of 2 Gaussian fit results. This is especially the case until the peak of cellular thermogenesis, when the correlation coefficients for all MTM fits are greater than 0.95. After this peak, the model deviates from the experimental results with correlation coefficients between 0.67 and 0.85 (Fig 5B and S2 Table). This deviation is likely caused by the model's assumptions. One assumption that leads to this deviation, at times greater than $t_{peak}$, is that of steady-state growth throughout the entire time domain of the model. This assumption is intrinsically accounted for in the MTM by the incorporation of the steady-state ATP flux calculated from the iML1515 model. The ATP flux must be maximal during steady-state growth and decrease as the cell transitions its metabolism to that of the stationary phase. Nevertheless, the agreement between the experimental and model values supports the argument that there is a chemical-level control of cellular thermogenesis.

To evaluate MTM's performance, the standard error of the estimate (SEE) was calculated for both the sum of 2 Gaussian (2G) equations fit to the observed data and the MTM's prediction (S2 Table). The Δ*arcB* SEE value for the MTM model compared to the experimentally observed data was 0.051, whereas that of the fitted 2G model for this strain's experimental data was 0.012. Thus, the MTM model predicts heat generation to similar errors as data specifically fit to experimentally observed data. Such an agreement between MTM and the sum of 2 Gaussian models supports the argument that ATP is the major, even if not the sole, contributor to cellular microbial thermogenesis.

Finally, the MTM was applied to predict the heat generation in the *glnL+* strain. The MTM showed a better agreement with knockout strains' heat generation than that of the *glnL* overexpression strain (S5 Fig and S2 Table). This disagreement is likely due to the limitations of the ATP assay measurements near or below detectable limits. Since the *glnL* overexpression strain has an increased lag phase, this imprecision becomes more important than the knockout or wild-type strains. This is because the *glnL* overexpression strain peaks in thermogenesis during the early log phase, whereas the kinase knockout and wild-type strains peak in late log phase, meaning that they get 1 clear reading of intracellular ATP before the peak, whereas the *glnL* overexpression strain may not.

## Discussion

Here, we present the protein kinases ArcB, GlnL, and YccC as the set of thermogenic regulators that are acting through intracellular ATP to modulate microbial thermogenesis. This fundamental study builds on previous work by extending it to engineer microbial systems for a variety of potential applications. We show this engineering potential for the first time through the overexpression of *glnL*. We envision leveraging the foundational understanding of intracellular ATP concentration's impact on cellular thermogenesis to design high-heat producing organisms. For example, probiotic creams supplemented with an essential nutrient cocktail for high-heat producing cells can be developed. These types of applications could find particular use in cold-water medicine and diving applications.

We propose that the putative mechanism that captures the coupled effects of cellular thermogenesis and ATP levels works through the ATP synthase complex. It has recently been recognized that the ATP synthase complex can act in reverse to hydrolyze cellular excesses of ATP in response to increased cellular ATP levels [59]. Thus, in the knockout strains presented, where we have an increase in cellular ATP concentrations, this reverse ATP synthase activity can capture the observed coupling of the peaks in the intracellular ATP level and microbial thermogenesis. This mechanism also explains the results of the *glnL* overexpression assay, as a decrease in heat generation is observed with a concomitant reduction in the intracellular ATP level (Fig 4). This type of mechanism, in which the cell balances increased intracellular

concentrations of ATP by reversal of ATP synthase activity, follows our kinetic intuition about cellular metabolism. An increase in intracellular ATP is assumed to increase the equilibrium driving force for all reactions involving ATP. This represents a vast and diverse set of enzyme-catalyzed metabolic reactions, with 1 key similarity—this class of the enzymes conserves a common metal cofactor, $Mg^{2+}$. The $Mg^{2+}$ limitation has already been shown to decrease the growth rates of microbial cells such as *E. coli* as this metal cofactor is involved in a wide variety of key cellular processes, including the stabilization of the cellular membrane [60]. Thus, the reverse activity of the ATP synthase complex allows the cell to respond to unusually high levels of ATP by hydrolyzing the intracellular excess ATP, to free $Mg^{2+}$ necessary for other key cellular processes.

Although this work supports important conclusions in the field, there are some limitations. For example, the results cannot be directly extended to mammalian cells. Mammalian cells have distinct regulatory mechanisms that differ greatly from those found in microbial systems [61]. However, ATP is a metabolite whose importance transcends the boundaries of evolutionary domains. Thus, it is likely that this metabolite plays a key role in mammalian thermogenesis as well at the biochemical level.

Further study is also needed to understand whether there are any metabolic enzymes, specifically those with ATP hydrolysis domains, which also regulate thermogenesis. Furthermore, there is immense value in understanding the exact biochemical pathways that influence cellular thermogenesis. The interplay between all of the proteins present in the signaling cascades of these autophosphorylating protein kinases complicates this understanding. Additionally, we need to discern the role that environmental signals can play in cellular heat generation. Finally, the rational design of other highly thermogenic microbes will also proceed from this work.

## Methods

### Screening rationale and strain selection

A screen of single-gene knockouts of *E. coli* K-12 BW25113 was conducted. A library of knockout strains, the Keio Collection, was purchased from Horizon Discovery (Waterbeach, United Kingdom) [46]. This library consists of single-gene knockouts of every non-essential gene in *E. coli* BW25113. This accounts for 3,985 genes out of the 4,453 genes present in the organism's genome. The experiments were run in an 8-channel TAM Air Calorimeter (TA Instruments; New Castle, Delaware, United States of America) using water as a negative control sample for each well, as is suggested for liquid samples in the instrument manual. For strain selection, a python script was developed to search the Keio collection for genes of interest. See S1 Note for details.

### Microbial thermogenesis assay

Six single-gene knockout strains were grown overnight along with 1 wild-type (positive control) and 1 media control (negative control) for a total of 8 samples with shaking at 37°C and 250 rpm in a 96-deep well plate. Samples are tested in singlicate ($N = 1$) for the initial screen. Then, if the sample presents >25% increase in thermogenesis, it was tested again ($N \geq 3$). For all experiments, cells were grown in Luria–Bertani media (Sigma Aldrich 71753–6) supplemented with 2% glucose (Sigma Aldrich G7021). Following overnight incubation, $OD_{600}$ (by $Abs_{600}$, absorbance at 600 nm) was measured in a clear bottom 96-well plate, and the culture was diluted to a final $OD_{600}$ of 0.005 in 10 mL growth media in a 20 mL glass ampoule (TA Instruments). The ampoule was then sealed and placed inside the TAM Air Calorimeter for 4 to 5 h for heat flow measurement. After all samples had reached their peak, they were removed from the calorimeter and again measured for their $OD_{600}$. See S2 Note for more details.

## Thermogenesis data analysis

Initially, the data for each experiment were plotted as heat flow (W) over time (s). These data were baseline-normalized by subtracting out the negative control value from each sample value at each time point (collection frequency of 0.1 Hz). Then, the data were normalized by final $OD_{600}$ respective to the experimental strain to yield the normalized heat flow (mW/OD). To understand how the heat generation compares across experiments, the fold change was calculated by dividing each normalized heat flow by the wild-type normalized heat flow of that experiment. Final $OD_{600}$ measurements were consistently taken within 30 min of the heat flow maximum. The calculation for significance was conducted by unpaired $T$ test, using Welch's correction, using GraphPad Prism 9. All plots were also constructed using GraphPad Prism 9.

## Microaerobic bacterial growth assay

Cells were inoculated into growth media and cultured overnight from frozen stock stored at −80°C. Overnight culture was then seeded to a starting $OD_{600}$ of 0.005 in 20 mL glass ampoules. Ampoules were capped and sealed at time point zero. Every 30 min, ampoules were uncapped, and $Abs_{600}$ measurements were taken in a clear-bottomed, black-walled, 96-well plate. These measurement values were converted to $OD_{600}$ by multiplying by 1.75, which was an experimentally determined conversion factor. All samples were tested in biological triplicate.

## Plasmid construction for the kinase overexpression

Gene coding sequences for *yccC*, *glnL*, and *arcB* (genome) were amplified from the genomic DNA of *E. coli* BW25113 strain by PCR. All gene sequences were from NCBI genome CP009685.1. The overexpression plasmid backbone was amplified and cloned by Gibson assembly as previously described [62]. All plasmid sequences were verified by DNA sequencing (GENEWIZ, South Plainfield, New Jersey, USA). All the oligonucleotides were purchased from Integrated DNA Technologies (IDT, Coralville, Iowa, USA). All plasmid minipreps were performed with PureLink Quick Plasmid Miniprep Kit (Invitrogen, Waltham, Massachusetts, USA). Once plasmid sequences were verified, they were transformed into *E. coli* BW25113 strain and induced with 0.4 ng/mL anhydrotetracycline (aTc), and the resultant strains were assayed for cellular thermogenesis.

## ATP quantification assay

ATP measurements were conducted in the following way. Cells were grown in 20 mL glass ampoules (TA instruments) in 10 mL of growth media. The ampoules were capped and sealed at time point zero. Every 30 min, ampoules were unsealed, and the intracellular ATP concentration was measured in triplicate for each strain (*ΔarcB*, *ΔglnL*, *ΔyccC*, and wild-type) as follows. The cells were first lysed, using boiling water prepared on a magnetic hot plate. From the ampoule, 1 mL of the sample was taken into a 1.5 mL centrifuge tube (containing about $10^6$ cells) and pelleted in a microcentrifuge at 12,000×g for 10 min. The Invitrogen ATP Determination Kit (A22066) was then used to quantify the ATP concentration. $OD_{600}$ measurements were also made at each time point and used to calculate an average intracellular ATP concentration assuming 1 $OD_{600}$ as $8 \times 10^8$ cells/mL and an average cell volume as 1 fL [63,64].

## Microbial thermogenesis model construction

The MTM was developed under 3 key assumptions: the sole metabolic determinant of microbial thermogenesis is ATP; the intracellular ATP flux scales with the intracellular ATP

concentration; the ATP flux is constant at its steady-state value throughout the thermogenesis assay. The growth, the intracellular ATP concentration, and thermogenesis raw data were fit using the Gompertz growth, lognormal, and sum of 2 Gaussian equations, respectively, to get analytic functions over time representing the data (S6–S8 Figs). All models demonstrated good agreement with raw data (S3 and S4 Tables). Model fitting and correlation coefficient calculations were conducted using Microsoft Excel and GraphPad Prism 9. A MATLAB algorithm was then developed to output a heat generation prediction (mW) from input growth, ATP concentration, and ATP flux parameters. The steady-state ATP metabolic flux parameters were calculated in a python script using constraint-based optimization of metabolite flux from the COBRApy package and iML1515 model [58]. In short, the MATLAB algorithm converts growth data into cell number data, which were then used to calculate the total intracellular ATP concentration-weighted ATP flux through each cell. Then, this flux was converted to a heat energy release for the entire culture.

## Model selection

Model equations for microbial growth, intracellular ATP concentration, and thermogenesis were chosen as follows. A priority of this modeling was to choose analytic functions of time for these models. For intracellular ATP fitting, the lognormal equation, which is frequently used to model biological metabolites, was utilized [65]. Model selections for microbial thermogenesis and growth were based on testing a range of different model types and selecting what has the best fit to the data. Goodness of fit was evaluated by replicates test with $\alpha = 0.05$ (S3 and S4 Tables and S9 and S10 Figs). The statistical analyses were conducted in GraphPad Prism 9. AUC analysis was conducted by integrating the continuous equations of intracellular ATP level and cellular thermogenesis using MATLAB 2022a.

## MTM evaluation

To evaluate MTM performance, an SEE analysis was conducted. SEE is calculated from the following equation:

$$SEE = \sqrt{\frac{\sum (y_{observed} - y_{predicted})^2}{n}},$$

where $y_{observed}$ are the measured microbial heat outputs (W), $y_{predicted}$ are the predicted microbial heat outputs (W), and $n$ is the total number of the sample datapoints.

## Statistics

Statistical analyses and regressions were performed using Prism 9 (GraphPad). Quantitative values were averaged and shown by their means and standard deviations (SDs). These values were tested for statistical significance of differences by two-tailed Student's $T$ test using Welch's correction, to account for differences in sample's SDs or ANOVA. Statistical significance is stated for samples with $p$-values lower than 0.05.

## Supporting information

**S1 Note. The script scans through a predefined list of target genes and parses through the Keio Collection library to locate the position of the user-defined genes of interest.** If a gene was not found, it was searched under its various pseudonyms from MetaCyc. If a gene could not be found, it was not tested. This was the case for 5 transcription factors. Five proteins listed as transcription factors for *E. coli* K-12 were unable to be found in the Keio Collection under

their listed name or any synonyms. These genes were: *mazE*, *mqsA*, *dicA*, *birA*, and *dnaA*. This is because these genes are essential to the *E. coli* K-12 BW25113 strain.
(DOCX)

**S2 Note. The media in this study were chosen to supply the necessary nutrients for core metabolism.** Supply of these nutrients meant, under the assumption that heat generation is due to catabolism of nutrients, that the ideal conditions for maximal heat generation were provided for cells. However, note that in the calorimeter, cells cannot be growing completely aerobically as there is no air flow into the ampoule, and the calorimeter does not allow for shaking, meaning poor aeration of media. Future experimentation will be aimed at understanding how the oxygen concentration changes with time in the ampoule and affects heat output. Additionally, a total of 449 experimental knockout strains were tested, representing 370 unique knockout strains. The number of experimental strains is greater than the unique tested strains due to replicate testing.
(DOCX)

**S1 Data. Individual replicate data from all quantitative analyses.** Each figure's individual data is recorded in its own tab of the Excel sheet. The Excel sheet is labeled: S1_Data.xlsx.
(XLSX)

**S1 Table. Raw data from the thermogenesis screen.** The raw values of the fold heat generation change over the wild-type value are shown in the attached Excel file. The range of the data is from 0.71 to 1.31. Gene knockouts are arranged in decreasing order of fold change and grouped by protein function. Genes in the "Other" category do not fall into another protein category or are not annotated with a function. The data table can be found in S1 Data.
(XLSX)

**S2 Table. Standard error of the MTM estimate.** The standard error of the estimates (SEE) of microbial thermogenesis of the sum of the 2 Gaussian equations (2G) and microbial thermogenesis model (MTM) are tabulated. Error from peak calculation takes the value of SEE and divides it by the average peak thermogenesis to understand the variance of the average peak thermogenesis.
(DOCX)

**S3 Table. Growth model selection.** A replicate test was used to test for model fit to the data. Comparison between 3 commonly used models of population growth of cells: Gompertz Growth, Malthusian Growth, and Exponential Plateau. The *p*-values and rejection of model decisions are shown as calculated.
(DOCX)

**S4 Table. Thermogenesis model selection.** Comparison between 7 different models for microbial heat generation curves. A replicate test was used to test for model fit to the data with *p*-values less than 0.01 defined as a rejection of the model. Statistical analyses were conducted in GraphPad Prism 9.
(DOCX)

**S1 Fig. Heat flow fold change of high-heat generating strains.** Bar plot of heat generation fold change normalized respective to the experimental wild-type value for high-heat generating strains. Individual experimental values are shown as points. *ΔarcB*–red, *ΔglnL*–blue, *ΔyccC*–orange, and wild-type–the dashed line. Data for all individual replicates can be found in S1 Data.
(TIF)

**S2 Fig. Area under the curve analysis.** Area under the curve (AUC) analysis was conducted on both intracellular ATP concentration (empty bars) and heat generation curves (striped bars). *ΔarcB*–red, *ΔglnL*–blue, *ΔyccC*–orange, and wild-type–black. Data for all individual replicates can be found in S1 Data.
(TIF)

**S3 Fig. Microbial thermogenesis curves for ArcB, GlnL, and YccC overexpression strains.** All 3 kinase overexpression strains were evaluated for heat generation (W) over time (h) with and without inducer aTc. Strains grown in the presence of the inducer are shown in solid lines, and uninduced samples are shown in dashed lines. ArcB overexpression (green), GlnL overexpression (purple), YccC overexpression (pink), WT (black), and blank medium negative control (gray). The $OD_{600}$ values are also shown to note the cell growth at the 4 h time point. Data for all individual replicates can be found in S1 Data.
(TIF)

**S4 Fig. AUC analysis of *glnL* overexpression's intracellular ATP and thermogenesis data.** Area under the curve (AUC) analysis was conducted on both intracellular ATP (empty bars) and heat generation curves (striped bars). AUC was conducted by integrating continuous models of the data with a bin size of 0.5 h as the lowest resolution of the measurement. Data for all individual replicates can be found in S1 Data.
(TIF)

**S5 Fig. MTM prediction of *glnL* overexpression.** Average heat generation is fitted by the sum of 2 Gaussian equations (solid line) and plotted against the respective MTM approximations for the strain's heat generation over time (dashed line). Data for all individual replicates can be found in S1 Data.
(TIF)

**S6 Fig. Thermogenic strains' growth curves.** Experimental microaerobic growth data for all strains ($OD_{600}$—solid line) fit with the Gompertz growth model equation (dashed). *ΔarcB*–red, *ΔglnL*–blue, *ΔyccC*–orange, and wild-type–black. Gompertz growth equation: a–asymptote at $t \to \infty$, b–displacement along x-axis, c–growth rate, and t–time. Error bars: ± standard deviation. $N = 4$ for all strains. Data for all individual replicates can be found in S1 Data.
(TIF)

**S7 Fig. Intracellular ATP concentration of thermogenic strains.** Experimental intracellular ATP concentration data (mM—solid line) fit with the lognormal model equation (dashed). $N = 2$ for all samples. The model equation for the lognormal function is shown. *ΔarcB*–red, *ΔglnL*–blue, *ΔyccC*–orange, and wild-type–black. Lognormal equation: M–geometric mean, S–geometric standard deviation, t–time, and A–factor related to amplitude. Error bars: ± SD. Data for all individual replicates can be found in S1 Data.
(TIF)

**S8 Fig. Heat generation curves of thermogenic strains.** Experimental thermogenic strain heat flow data (mW—solid line) fit with the 2 Gaussian equations (dashed). The model equation for the sum of 2 Gaussian functions is shown. *ΔarcB*–red, *ΔglnL*–blue, *ΔyccC*–orange, and wild-type–black. Sum of 2 ($i = 1,2$) Gaussian equations: A–amplitude, $\bar{X}_i$–mean, t–time, and $\sigma_i$–standard deviation. Data for all individual replicates can be found in S1 Data.
(TIF)

**S9 Fig. Bacterial growth model selection.** Various regression model equations (dashed line) were fit to the experimental data for microbial growth–G ($OD_{600}$–solid line). (A) Gompertz

Model: $Y_M$—Maximum population, $Y_0$—initial population, k–rate constant, and t–time. (B) Malthusian Model: $Y_0$—initial population, and k–rate constant. (C) Exponential Plateau: $Y_M$—Maximum population, $Y_0$—initial population, k–rate constant, and t–time. $\Delta arcB$–red, $\Delta glnL$–blue, $\Delta yccC$–orange, and wild-type–black. Error bars: ± SD. Data for all individual replicates can be found in S1 Data.
(TIF)

**S10 Fig. Heat generation model selection.** Various regression model equations (dashed line) were fit to the experimental data (W–solid line) for microbial heat generation (HG). (A) Single Gaussian Model: A–amplitude, $\bar{X}$–mean, t–time, and $\sigma$– standard deviation. (B) Lognormal Model: M–geometric mean, S–geometric standard deviation, t–time, and A–factor related to amplitude. (C) First order polynomial: $B_i$ – coefficient, and t–time. (D) Second order polynomial: $B_i$ – coefficient, and t–time, (E) Third order polynomial: $B_i$ – coefficient, and t–time, (F) Fourth order polynomial: $B_i$ – coefficient, and t–time. $\Delta arcB$–red, $\Delta glnL$–blue, $\Delta yccC$–orange, and wild-type–black. Data for all individual replicates can be found in S1 Data.
(TIF)

**S11 Fig. Experimental microbial thermogenesis including error bars.** Average strain heat flow (mW) curves over time. All samples were tested in at least triplicate, with distinct samples ($N > 3$). $\Delta arcB$–red, $\Delta glnL$–blue, $\Delta yccC$–orange, and wild-type–black. Error bars: ± SD. Data for all individual replicates can be found in S1 Data.
(TIF)

**S12 Fig. Microbial thermogenesis of *glnL* overexpression strain including error bars.** Average strain heat flow (mW) curves over time. All samples were tested in triplicate, with distinct samples. *glnL*+ ($N = 3$, purple) and wild-type ($N = 3$, black). Error bars: ± SD. Data for all individual replicates can be found in S1 Data.
(TIF)

## Acknowledgments

The authors thank Dr. Jinjin Diao for his comments and feedback on the manuscript.

The content is solely the responsibility of the authors and does not necessarily represent the official views of the funding agency.

## Author Contributions

**Conceptualization:** Tae Seok Moon.

**Data curation:** Puneet Singh Dhatt, Tae Seok Moon.

**Formal analysis:** Puneet Singh Dhatt, Tae Seok Moon.

**Funding acquisition:** Tae Seok Moon.

**Investigation:** Puneet Singh Dhatt, Stephen Chiu, Tae Seok Moon.

**Methodology:** Puneet Singh Dhatt, Tae Seok Moon.

**Project administration:** Tae Seok Moon.

**Resources:** Tae Seok Moon.

**Supervision:** Tae Seok Moon.

**Validation:** Puneet Singh Dhatt, Stephen Chiu, Tae Seok Moon.

**Visualization:** Puneet Singh Dhatt.

**Writing – original draft:** Puneet Singh Dhatt, Tae Seok Moon.

**Writing – review & editing:** Puneet Singh Dhatt, Stephen Chiu, Tae Seok Moon.

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
