## [Editor Report · Decision Letter 0]

30 May 2023

Dear Dr. Moon, 

Thank you for submitting your manuscript entitled "Elucidating the mechanism of microbial thermogenesis" for consideration as a Research Article by PLOS Biology.

Your manuscript has now been evaluated by the PLOS Biology editorial staff, as well as by an academic editor with relevant expertise, and I am writing to let you know that we would like to send your submission out for external peer review.

Once your full submission is complete, your paper will undergo a series of checks in preparation for peer review. After your manuscript has passed the checks it will be sent out for review. To provide the metadata for your submission, please Login to Editorial Manager (https://www.editorialmanager.com/pbiology) within two working days, i.e. by Jun 01 2023 11:59PM.

Kind regards,

Paula

---

Senior Editor

PLOS Biology

---

## [Decision Letter · Decision Letter 1]

13 Jul 2023

Dear Dr. Moon,

Thank you for your patience while your manuscript "Elucidating the mechanism of microbial thermogenesis" went through peer-review at PLOS Biology. Your manuscript has now been evaluated by the PLOS Biology editors, an Academic Editor with relevant expertise, and by several independent reviewers.

In light of the reviews, which you will find at the end of this email, we are pleased to offer you the opportunity to address the comments from the reviewers in a revision that we anticipate should not take you very long. We will then assess your revised manuscript and your response to the reviewers' comments with our Academic Editor aiming to avoid further rounds of peer-review, although might need to consult with the reviewers, depending on the nature of the revisions.

We consider that the experiments suggested by reviewer #2 are needed for further consideration. Please address all the reviewers' issues. 

**IMPORTANT - SUBMITTING YOUR REVISION**

*Resubmission Checklist*

*Published Peer Review*

*PLOS Data Policy*

*Blot and Gel Data Policy*

Sincerely,

Paula

---

Senior Editor

PLOS Biology

REVIEWS:

Reviewer #1: Human microbiome and heat generation.

Reviewer #2: Synthetic biology.

Reviewer #1: Soon and colleagues are describing some mechanisms for heat generation within microbes.

Through a knock out screen, three self-phosphorylating protein kinases were identified to provide extreme thermogenic properties.

This presents a very interesting scientific result which may lead to very interesting applications.

#major comments

An expanded discussion of the potencial mechanisms integrating the results of the genetic screen, experimental data and simulations.

Fig.3. It would be very informative to display the variability on the times series of growth, intracellular ATP and heat generation

#minor comments

Fig. 2 A is not readable. Complete listings could be send to supplementary with activity values.

Fig. 2B maybe could be expanded to separe the four different families of mutants

Label in Fig. 4 is a bit misleading. I would not use "validation". I think it is more appropriate to talk about Microbial thermogenesis model analysis.

MTM model would benefit from more expanded information. Why lognormal was selected? Why the sum of two lognormal? Was it after comparing testing different types of functions? Correlation coefficient are associated with linear model. I wonder how correlation coefficient is here used to describe the model fit.

Reviewer #2: Dhatt and Seok Moon present a screen for molecular determinants of thermogenesis in laboratory strains of Esherichia coli. Using a set of knockout strains from the Keio collection, the authors assess heat generation during logarithmic growth and identify 3 protein kinases whose deletions dysregulates thermogenesis. Further, they correlate intracellular ATP levels to heat generation. Altogether, the authors present an interesting dive into the molecular mechanisms that govern heat generation during bacterial growth. I believe the manuscript is suitable to publication with some minor additional text edits, analysis, and experiments:

1. The connection between thermogenesis and the microbiome is quite farfetched, especially given the low abundance of E. coli in the human microbiome, the vastly altered metabolic state of cells in the gut vs the lab, and the lack of generalizability of the findings to other microbial taxa that dominate the host-associated microbiomes. I would suggest incorporating an expanded and more nuanced discussion if the authors intend to extend these findings to complex microbial communities. 

2. The model of how the three identified protein kinases play into thermogenesis could be clearer. Unless I am misunderstanding, deletion of any of the kinases leads to increase levels of intracellular ATP, which is correlated with increased heat generation. Since these proteins are deleted, their enzymatic activity on ATP plays no role in heat generation in the deletion mutants, suggesting that these kinases are rather low heat spenders of ATP. Is the model then that ATP can be acted upon by other cellular processes that are more exothermic than kinase activity? If so, what would these processes be? 

3. The link between ATP hydrolysis could be strengthen by testing conditions wherein intracellular levels of ATP are lowered with respect to wild-type bacteria. Would overexpression of these protein kinases deplete ATP and perhaps lower heat generation? Are there other means to engineer a system with lower ATP concentrations to examine the effect on heat generation?

4. In Figure 3, the correlation between ATP and heat is made purely by eye. Could the authors include a more quantitative comparison, perhaps by correlating the AUC of ATP to heat generation in a scatter plot?

---

## [Editor Report · Decision Letter 2]

8 Sep 2023

Dear Dr. Moon,

Thank you for your patience while we considered your revised manuscript "Elucidating the mechanism of microbial thermogenesis" for publication as a Research Article at PLOS Biology. This revised version of your manuscript has been evaluated by the PLOS Biology editors, the Academic Editor [and the original reviewers -EDIT AS APPLICABLE].

Based on our Academic Editor's assessment of your revision, we are likely to accept this manuscript for publication, provided you satisfactorily address following data and other policy-related requests.

1. DATA POLICY:

A) Supplementary files (e.g., excel). Please ensure that all data files are uploaded as 'Supporting Information' and are invariably referred to (in the manuscript, figure legends, and the Description field when uploading your files) using the following format verbatim: S1 Data, S2 Data, etc. Multiple panels of a single or even several figures can be included as multiple sheets in one excel file that is saved using exactly the following convention: S1_Data.xlsx (using an underscore).

B) Deposition in a publicly available repository. Please also provide the accession code or a reviewer link so that we may view your data before publication.

Regardless of the method selected, please ensure that you provide the individual numerical values that underlie the summary data displayed in the following figure panels as they are essential for readers to assess your analysis and to reproduce it: Figures 2ABC, 3ABC, 4ABCD, 5B, Supplementary figures S1, S2, S3, S4, S5, S6, S7, S8, S9ABC, S10ABCDEF, S11, S12.

**Please also ensure that figure legends in your manuscript include information on where the underlying data can be found, and ensure your supplemental data file/s has a legend.**

2. We suggest a change in the title: "Microbial thermogenesis is dependent on ATP concentrations and the protein kinases ArcB, GlnL and YccC"

We expect to receive your revised manuscript within two weeks.

*Published Peer Review History*

*Press*

Sincerely,

Paula

---

Senior Editor,

pjaureguionieva@plos.org,

PLOS Biology

---

## [Editor Report · Decision Letter 3]

14 Sep 2023

Dear Dr. Moon,

Thank you for the submission of your revised Research Article "Microbial thermogenesis is dependent on ATP concentrations and the protein kinases ArcB, GlnL, and YccC" for publication in PLOS Biology. On behalf of my colleagues and the Academic Editor, Matthew Wook Chang, I am pleased to say that we can in principle accept your manuscript for publication, provided you address any remaining formatting and reporting issues. These will be detailed in an email you should receive within 2-3 business days from our colleagues in the journal operations team; no action is required from you until then. Please note that we will not be able to formally accept your manuscript and schedule it for publication until you have completed any requested changes.

PRESS

Sincerely, 

Paula 

---

Senior Editor

PLOS Biology
